# Brain-inspired Multi-View Incremental Learning for Knowledge Transfer and Retention

## Abstract

The human brain exhibits remarkable proficiency in dynamic learning and adaptation, seamlessly integrating prior knowledge with new information, thereby enabling flexible memory retention and efficient transfer across multiple views. In contrast, traditional multi-view learning methods are predominantly designed for static and fixed-view datasets, leading to the notorious "view forgetting phenomenon", where the introduction of new views leads to the erosion of prior knowledge. This phenomenon starkly contrasts with the brain's remarkable ability to continuously integrate and migrate past knowledge, ensuring both the retention of old information and the assimilation of new insights. This oversight presents a critical challenge: how to efficiently learn and integrate new views while simultaneously preserving knowledge from previously acquired views and enabling flexible knowledge transfer across diverse views. Inspired by underlying neural processing mechanisms, we propose a view transfer learning framework named Hebbian View Orthogonal Projection (HVOP), which realizes efficient knowledge migration and sharing between multi-view data. HVOP constructs a knowledge transfer space (KTS), where the KTS reduces the interference between the old and the new views through an orthogonal learning mechanism. By further incorporating recursive lateral connections and Hebbian learning, the proposed model endows the learning process with brain-like dynamic adaptability, enhancing knowledge transfer and integration, and bringing the model closer to human cognition. We extensively validate the proposed model on node classification tasks and demonstrate its superior performance in knowledge retention and transfer compared to traditional methods. Our results underscore the potential of biologically inspired mechanisms in advancing multi-view learning and mitigating the view forgetting phenomenon.

## 1 Introduction

Brain-inspired computing seeks to mimic the human brain's ability to process information through its complex network of neurons and synapses, enabling remarkable capabilities in learning and memory retention. Neuroscientific research reveals that the brain can continually adapt and reconfigure its neural connections in response to new stimuli Deco et al. (2011); Hassabis et al. (2017). This dynamic capability underlies cognitive functions such as learning from multiple sensory inputs and retaining memories over time Stein & Stanford (2008).

The need for such adaptability is particularly evident in modern computational environments where we increasingly deal with incremental views, such as in medical image analysis Konz & Mazurowski (2024); Weng et al. (2024); Zhou et al. (2023), social network analysis Meng et al. (2024); Wen et al. (2023); Song et al. (2023), recommendation systems Li et al. (2024b); Paliwal et al. (2024); Prakash et al. (2023), and video surveillance Bao et al. (2022); Liu et al. (2020). In these applications, new data sources, or "views", continuously emerge, challenging traditional multi-view learning methods that were originally designed for static datasets. These traditional methods Zhao et al. (2017); Wu et al. (2023) struggle with dynamic data integration, lacking the brain-like flexibility to adapt to new views without forgetting previously acquired knowledge. This necessitates costly retraining to integrate new views or risks discarding previously learned insights when trained directly on new data. Such static approaches starkly contrast with the brain's fluid and adaptive learning processes, underscoring a significant gap in existing methodologies.

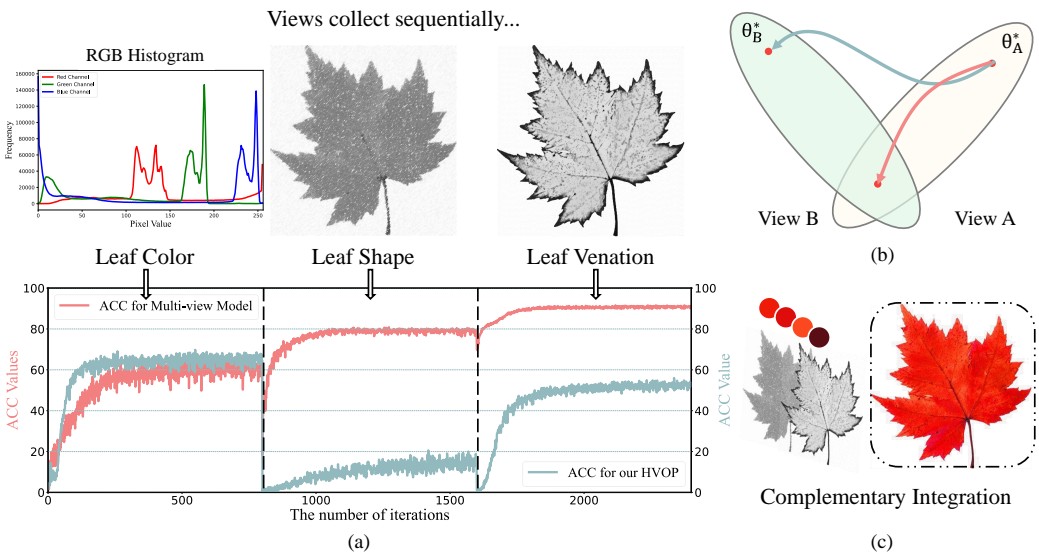

Figure 1: (a) Traditional multi-view methods suffer from view forgetting in the view increment setting in dataset 100leaves. Leaf color, shape and venation views are sequentially entered into the model for learning, and the performance of the model depends only on the quality of the current view. (b) Plot of the updated direction of weights for the arrival of the new view, where the blue curve represents the updated direction of the traditional method and the red curve represents the desired updated direction. (c) The graph of view complementarity, where the combination of leaf shape, venation, and color produces the most complete representation of the data.

As illustrated in Fig.2(a), the performance of traditional models with settings of view increments (marked in blue) deteriorates because these models fail to effectively transfer the knowledge accumulated from previous views to assist in learning the new view. Instead, they rely solely on the information from the new view to complete the learning process independently. Such a learning pattern is heavily constrained by the specific performance of the current learned view, limiting the model's overall knowledge integration and generalization capabilities. The gradient direction of the old view is forgotten during the weight updating process and is directly docked onto the new view (Fig.2(b) blue line) without keeping the old and new views updated in the same direction (Fig.2(b) red line)Kirkpatrick et al. (2017). Moreover, the relationships between new and old views, such as consistency (i.e., similarity in descriptions of the same object or event across different views) and complementarity (i.e., different views provide unique and mutually complementary information) (as shown in Fig. 2(c)), require the model to possess strong knowledge integration capabilities to efficiently assimilate the incremental information from the new view while maintaining the coherence of knowledge from the old view.

To address these challenges, it is instructive to consider how the human brain manages similar tasks. The brain's ability to handle information from various sensory inputs without losing prior knowledge is underpinned by its unique structure and function Chiel & Beer (1997); Ito et al. (2022). Notably, the hippocampus plays a crucial role in this process, aiding in the formation and storage of new memories as well as in the retrieval of existing memories without interference from newly incoming information. This capability is achieved through sophisticated mechanisms, such as the creation of separate but interconnected neural pathways for new and old memories, allowing for the simultaneous retention of stability and plasticity in neural representations Kumaran et al. (2016); McClelland et al. (1995); Scoville & Milner (1957); Squire (1992). These insights into hippocampal function suggest a model of memory that is both dynamic and robust, essential characteristics for the development of effective multi-view learning algorithms.

To bridge the gap between human cognitive abilities and machine learning models, we propose a novel framework that draws direct inspiration from neural mechanisms. Our model, termed Hebbian View Orthogonal Projection (HVOP), integrates concepts from neuroscience to enhance the

learning architecture's capacity for knowledge retention and transfer in multi-view learning tasks. Specifically, the framework employs mechanisms analogous to:

- **Lateral connections in neural circuits:** These are simulated in our network architecture to facilitate the integration of information across different views, enhancing the model's ability to maintain a coherent representation of the data Harris & Mrsic-Flogel (2013).

- **Synaptic plasticity via Hebbian learning:** This is used to adapt the synaptic weights, ensuring that the connections are strengthened or weakened according to their utility in task performance Feldman (2012); Gerstner et al. (2018). This adjustment supports the mechanism of orthogonal projection.

- **Memory retention and transfer in the hippocampus:** We mimic this aspect by introducing a specific space within our model – the Knowledge Transfer Space (KTS). This is akin to regions in the hippocampus that are responsible for encoding new memories and consolidating long-term ones, respectively Cohen (1993).

This brain-inspired approach not only addresses the key limitations of existing multi-view learning models but also provides a more robust framework for handling dynamic, incrementally available data. By employing lateral connections and Hebbian learning, our methodology approximates orthogonal projection, a crucial function for minimizing information loss during the learning of evolving views. By embedding KTS into the learning process, HVOP demonstrates superior performance in knowledge retention and transfer, effectively mimicking the human brain's capacity to integrate and preserve information across multiple sensory channels. The effectiveness of this approach is validated through extensive experiments across various multi-view datasets, where HVOP consistently outperforms both traditional and state-of-the-art multi-view learning methods.

## 2 RELATED WORK

### 2.1 MULTI-VIEW CONTINUAL LEARNING

Multi-view learning Wang et al. (2021); Yao et al. (2024); Guo et al. (2025) involves integrating and encoding information from multiple sources to derive a low-dimensional representation that captures both consistency and complementarity across views. Current multi-view continual learning has two dominant classifications: task-incremental and class-incremental types Van de Ven et al. (2022). Multi-view task-incremental learning Li et al. (2017); Sun et al. (2018) establish connections between tasks and extract consistent information across multiple views to complement each task learning. Thus, the multi-view class-incremental learning Yang et al. (2022); Qian et al. (2023); Li et al. (2024a) has been proposed, where new class data is seamlessly introduced without explicit prompting to the model, requiring the model to autonomously discern the onset of a new task. These methods focus regularizing the update of weights, thus alleviating the catastrophic forgetting. Catastrophic forgetting is the phenomenon in which a neural network forgets what it has learnt previously when learning a new task Ramasesh et al. (2021); De Lange et al. (2021); Elsayed & Mahmood (2024). Related research has focused on two strategies: knowledge retention and knowledge transfer. Knowledge retention methods help models retain information from old tasks while training new tasks by introducing memory mechanisms or employing regularization techniques Babakniya et al. (2024); Chen et al. (2021a;b). On the other hand, knowledge transfer methods attempt to transfer knowledge from a previous task to a new task, e.g., by using the output of the old task as a guide in the new task through knowledge distillation techniques Kang et al. (2023); Kumari et al. (2022); Zhou & Cao (2021). Despite ongoing research efforts to mitigate catastrophic forgetting, achieving a balance between retaining old knowledge and acquiring new knowledge remains a challenge. Our proposed view incremental learning is quite different from both task and class incremental learning settings. Notably, it operates without the need for prompts signaling new tasks, placing special emphasis on monitoring the transferability of knowledge across different views. This approach underscores the importance of capturing overall cognition, striving for equilibrium between preserving past knowledge and assimilating novel information.

## 2.2 Transfer Learning

Transfer learning is a pivotal method in machine learning to enhance model's efficiency by applying knowledge from previously learned tasks to new, related ones Falk et al. (2023); Lin & Reimherr (2024). It mirrors the way human brains utilize past experiences to learn new tasks faster. Traditionally, machine learning models are trained from scratch for each new task, which can be data-intensive and time-consuming. Transfer learning mitigates these challenges by reusing pre-trained models, thereby reducing the necessity for lengthy training periods. This technique has proven effective in diverse applications such as natural language processing Zhao et al. (2024); Ge et al. (2024), where models like BERT and GPT are adapted for tasks like sentiment analysis, and in computer vision Sohn et al. (2023); Jain et al. (2023), where convolutional neural networks pre-trained on large datasets are tailored for image recognition tasks. It also aids in adapting general speech recognition models to recognize specialized vocabulary or accents Kheddar et al. (2024); Lei et al. (2023). In multi-view learning, transfer learning is particularly crucial as it addresses the challenge of effectively transferring and utilizing knowledge across different views that may have varying distributions or feature spaces. Traditional multi-view transfer learning methods, as discussed in Chen et al. (2024); Tang et al. (2021), emphasize knowledge transfer across domains but often neglect the heterogeneity between views, especially when the views are not only different but also dynamically evolving. This oversight has spurred us to explore further and accomplish knowledge transfer among diverse views. In multi-view learning, transfer learning acts as a bridge, our work aims to achieve more effective knowledge transfer and integration within dynamically evolving view data.

## 3 Methodology

Drawing inspiration from the biological processes in the human brain, our methodology, Hebbian View Orthogonal Projection (HVOP), aims to replicate the dynamic adaptability seen in natural neural systems. This approach not only facilitates the integration and retention of knowledge across multiple views but also embodies the principles of synaptic plasticity and memory consolidation observed in biological networks.

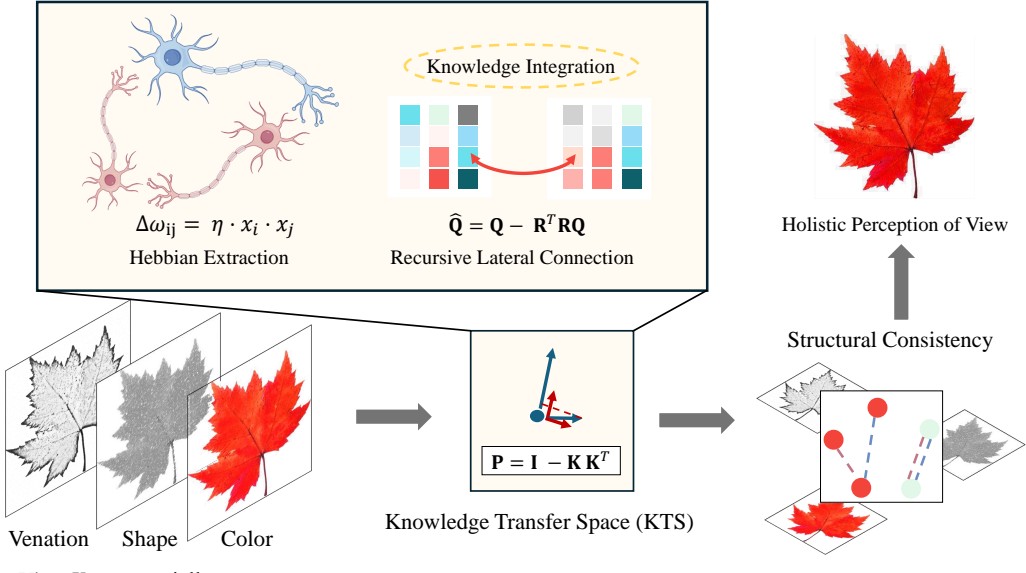

Figure 2: An overview of the proposed framework HVOP. HVOP refines a knowledge transfer space through recursive lateral connections and Hebbian learning, aiming at the transfer of view knowledge, while capturing the topological consistency of the multi-view structure, thus realizing holistic cognition in scenarios with incremental views.

### 3.1 Cause of the View Forgetting

Firstly we introduce the basic notations used in our framework. Denote given multi-view feature matrices as $\mathcal{X} = \{\mathbf{X}_v\}_{v=1}^{V}$, where $\mathbf{X}_v = [\mathbf{x}_1, \cdots, \mathbf{x}_n] \in \mathbb{R}^{n \times d_v}$ is the data from the $v$th view for any $v \in \{1, \cdots, V\}$. Here, $n$ is the number of samples, $V$ is the number of views, and $d_v$ denotes the distinct dimensionality of the feature matrices for each view. Additionally, $c$ represents the number of classes. We examine the phenomenon of view forgetting using $\mathcal{F}_{i,j}$, which denotes the testing accuracy of task $j$ after learning task $i$.

**Definition 3.1** *View Forgetting. Given a sequence of view data $\mathcal{X} = \{\mathbf{X}_v\}_{v=1}^{V}$, after learning the $v$ view data $\mathbf{X}_v$, the performance of the learned view $v$ is tested, represented as $\mathcal{F}_{v,v}$. When the learning of the last view data $\mathbf{X}_V$ is completed, the performance of the model significantly decreases in the previously tested views, shown as $\mathcal{F}_{v,V} < \mathcal{F}_{v,v}$.*

**Definition 3.2** *View Transfer Learning. Given a sequence of view data $\mathcal{X} = \{\mathbf{X}_v\}_{v=1}^{V}$, where the distribution of each view $\{\mathcal{D}_v\}_{v=1}^{V}$ differs significantly, but the task $\mathcal{T}$ remains the same. Transfer learning is designed to help improve the learning of the target prediction function $f(\cdot)$, which enables the model to achieve robust overall cognition despite the evolving data distributions.*

View forgetting occurs when the model encounters new views. Although the model initially captures the previous view, it struggles to retain this memory throughout the learning process. We attribute the core cause of this phenomenon to the dynamic changes in neural network parameters during training. We have conducted an in-depth analysis of the network's specific updating mechanisms. Our findings reveal that the gradient update direction is influenced by the entire input vector space. The transformation of view information shifts this gradient direction, leading to the forgetting of previously learned view information. Furthermore, in graph networks, weight updates depend not only on the feature gradients of individual nodes but also on the gradients of neighboring nodes and the structural gradients of the graph. As view data evolves, the graph's structural information changes, further complicating the retention of knowledge from earlier views. Notably, preventing forgetting is a prerequisite for effective knowledge migration. It is due to the fact that shifts in the direction of the gradient can impede this migration, making retention of learnt knowledge more challenging.

### 3.2 Model initialization on arrival of the first view

Firstly, we design a streaming graph learning model wherein upon the arrival of new view data $\mathbf{X}_v$, we construct the corresponding adjacency $\mathbf{A}_v$ by applying $k$-Nearest Neighbor ($k$NN). We utilize a shared weights single layer GCN, which can be represented as,

$$\mathbf{Z}_v = \mathcal{G}C(\mathbf{A}_v, \mathbf{X}_v) = \sigma(\hat{\mathbf{A}}_v \mathbf{X}_v \mathbf{W}). \tag{1}$$

$\hat{\mathbf{A}}_v = \widetilde{\mathbf{D}}_1^{-\frac{1}{2}} \widetilde{\mathbf{A}}_v \widetilde{\mathbf{D}}_v^{-\frac{1}{2}}$ and $\widetilde{\mathbf{A}}_v = \mathbf{A}_v + \mathbf{I}$. $\mathbf{A}_v \in \mathbb{R}^{n \times n}$ corresponds to an adjacency matrix constructed by $\mathbf{X}_v$. Herein, $\mathbf{W} \in \mathbb{R}^{d_v \times d}$ denotes the learnable shared weight matrix, which would be shared by all the view. Immediately after that, we designed a fully connected layer for implementing the classification, which can be represented as,

$$\mathbf{H}_v = \mathcal{F}C_{\mathbf{W}_f}(\mathbf{Z}_v), \tag{2}$$

where $\mathbf{H}_V \in \mathbb{R}^{n \times c}$ defaults to the final predicted output. Previous work has shown that the above models perform well in static multi-view learning scenarios. However, dynamic view-level learning has been neglected, especially when only a portion of the view data is captured during training, and the model is unable to reuse this information when posing the newly captured views. Therefore, the question is *how to realize view incremental learning without compromising performance?*

### 3.3 Introduction of the orthogonal projection

For the initial complete first-view data, we employ graph convolutional neural networks and fully connected layers to capture view-specific knowledge. For the continuous addition of subsequent view information, we aim to achieve two objectives: preserving existing knowledge and integrating

newly introduced knowledge. Specifically, the prior view knowledge must not be forgotten, and learning from new views should not interfere with the knowledge acquired from previous ones.

Assume the input vectors $\mathbf{x}_{old} \in \mathbb{R}^n$ from prior views, span the subspace $\mathbf{X}$. Considering that changes in view information affect the updating of weights, which leads to view forgetting, in order to preserve the knowledge of the old view, we need to impose restrictions on the direction of the gradient $\Delta\mathbf{W}^P$ update for the next view, which makes it orthogonal to the subspace $\mathbf{X}$, satisfactory that $\Delta\mathbf{W}^P\mathbf{x}_{old} = 0$, which $\mathbf{x}_{old}$ denotes the past input vectors. This ensures that the gradient update of the weights does not affect the integration of knowledge of past views, demonstrated as

$$(\mathbf{W} + \Delta\mathbf{W}^P)\mathbf{x}_{old} = \mathbf{W}\mathbf{x}_{old}. \tag{3}$$

To address this, we specifically designate the principal subspace of $\mathbf{X}$ as the **Knowledge Transfer Space (KTS)**. The KTS is pivotal in assimilating crucial information that enhances a global cognitive perspective, essential for cultivating a comprehensive understanding and insight across various domains.

If we calculate a projection matrix $\mathbf{P}$ to the subspace orthogonal to the KTS, then gradients can be projected as $\Delta\mathbf{W}^P = \Delta\mathbf{W}\mathbf{P}$. Inspired by Saha et al., for the gradient update of the fully connected layer, it can be adjust by subtracting its projection onto the space spanned by the top $k$ principal components,

$$\nabla_{\mathbf{W}}\mathcal{L} = \nabla_{\mathbf{W}}\mathcal{L} - \mathbf{K}\mathbf{K}^T\nabla_{\mathbf{W}}\mathcal{L} = (\mathbf{I} - \mathbf{K}\mathbf{K}^T)\nabla_{\mathbf{W}}\mathcal{L} = \mathbf{P}\nabla_{\mathbf{W}}\mathcal{L}. \tag{4}$$

Here, $\nabla_{\mathbf{W}}\mathcal{L}$ is the gradient of loss with respect to weight $\mathbf{W}$, $\mathbf{K}$ denotes the matrix of top $k$ principal components of representation calculated by SVD with a small batch of data and $\mathbf{K}\mathbf{K}^T$ is the projection matrix to the principal subspace of the input feature. This effectively filters out components of the gradient that align with the principal directions, potentially reducing interference from predominant features that are already well-represented in the model. It removes the part of the gradient of the new view that is relevant to the old view from the gradient of the new view, ensuring that the learning of the new view does not interfere with the learning outcomes of the old view. However, principal components $\mathbf{K}$ are mostly extracted by performing an SVD decomposition of the representation matrix, which is not able to capture important features dynamically.

### 3.4 Simulation of orthogonal projection by recursive lateral connections

In biological neural networks, recursive lateral connections are essential for integrating sensory information and higher-level processing. It mirrors the biological processes of synaptic plasticity, where the brain adjusts its connections based on new stimuli without overriding prior learning. Thus, we incorporate a recursive lateral connection mechanism and Hebbian learning within the Knowledge Transfer Space (KTS) to facilitate the extraction and integration of new knowledge. More specifically, we project the input features into the neural subspace and recursively compute them according to the following formula:

$$\mathbf{Q}^- = -\mathbf{R}^T\mathbf{O}, \tag{5}$$

where $\mathbf{R}$ encodes transformations between the new and prior knowledge, $\mathbf{O} = \mathbf{R}\mathbf{Q}$ is the space obtained by matrix projection of $\mathbf{Q}$, which stands for the integration of knowledge after lateral connections and $\mathbf{Q}^-$ is the space after recurrent mapping. We consolidate lateral connections and preparatory knowledge to obtain high-value bio-filtered information $\hat{\mathbf{Q}}$.

$$\hat{\mathbf{Q}} = \mathbf{Q} + \mathbf{Q}^- = \mathbf{Q} - \mathbf{R}^T\mathbf{R}\mathbf{Q} = (\mathbf{I} - \mathbf{R}^T\mathbf{R})\mathbf{Q}. \tag{6}$$

In this way, the new view knowledge in the KTS can be further integrated and enhanced to provide richer information for global cognition. Here, the $\mathbf{Q}$ will change when each view is added, and will spontaneously integrate the knowledge left over from the previous step of the model, represented as

$$\mathbf{Q}_v = \sigma(\hat{\mathbf{A}}_v\mathbf{X}_v\mathbf{W}) + \alpha\mathbf{Z}_{v-1}. \tag{7}$$

Among them, $\mathbf{Q}_v$ refers to the representation we fuse after each new view arrives, which is composed of the representation of the current view and the knowledge of the previous view. From Equation 6, we can conclude that the projection matrix $\mathbf{P}'$ as

$$\mathbf{P}' = \mathbf{I} - \mathbf{R}^T\mathbf{R}. \tag{8}$$

We find a striking similarity between equation 8 and the orthogonal projection mechanism in equation 4. Thus, as long as the recursive lateral connections $\mathbf{R}$ are equal or similar to the principal component matrix $\mathbf{K}$, the model can realize orthogonal projections, which inspires us to employ Hebbian learning to dynamically extract the principal component matrix. Oja rule, as an improvement of Hebbian learning, gradually approximates the principal direction of the data by adjusting the weight matrix. Its update formula is:

$$\mathbf{R}_{t+1} = \mathbf{R}_t + \eta(x_t y_t^T - y_t y_t^T \mathbf{R}_t), \tag{9}$$

where $\mathbf{R}_t$ denotes the dynamic principal matrix, $\eta$ refers to learning rate, $x_t$ is the input vector and $y_t$ indicates the output vector computed from the current weight matrix and inputs.

With this rule, the data is projected and normalized to gradually approach its principal direction, ultimately causing the weight matrix to approximate the principal components of the input data. This adaptive weight updating mechanism ensures that the new gradient directions do not interfere with the learned knowledge, thus replacing the principal component matrix originally generated by the SVD decomposition. In this way, we leverage the biological mechanisms of recursive lateral connections and Hebbian learning to achieve orthogonal projection and facilitate knowledge transfer within a view incremental framework.

To further ensure the structural consistency of the model when dealing with different views, we design the following loss function to ensure that the graph structural information can be preserved in the incremental learning of views: $\mathcal{L}_{RE} = \frac{1}{2} \sum_{t=1}^{|\mathcal{T}|} \|\mathbf{A}_t - \sigma\left(\mathbf{Q}_{|\mathcal{T}|} \cdot \mathbf{Q}_{|\mathcal{T}|}^T\right)\|_F^2$, where $\mathbf{A}_t$ denotes the adjacency in the past view information and $|\mathcal{T}|$ is the current training number of views. Despite the fact that the view data is in a constant state of flux, the intrinsic topology of the graph data shows surprising stability. With such constraints, we can capture the consistent representation between view information very well. Besides, for semi-supervised node classification, we calculate CrossEntropy and update parameters in HGE-DED, as follows $\mathcal{L}_{CE} = -\sum_{i \in \Omega} \sum_{j=1}^{c} \hat{y}_{ij} \ln y_{ij}$, where $\Omega$ refers to the set of labeled samples, $\hat{y}_{ij}$ usually denotes to one-hot encoding format and $c$ is the amount of classes. Our approach constantly transfers and retains knowledge in the view incremental setting, while approaching a more comprehensive overall perception.

## 4 Experiments

To further validate the effectiveness of the proposed method, we have designed the experimental section intended to answer the following key evaluation questions (EQs):

- **EQ1** Does HVOP achieve superior performance compared to its competitors for the semi-supervised multi-view classification task?
- **EQ2** Does HVOP successfully implement orthogonal projection to alleviate view forgetting in view incremental learning?
- **EQ3** Does HVOP capture the tight association between old and new views when new ones add up, and does it promote learning about the whole knowledge?

### 4.1 Excellent Overall Perception: Comparison to SOTA (EQ1)

We next evaluate the effectiveness of our method on the node classification task compared with several classical and state-of-the-art methods in Table 1, where the best performance is highlighted in bold and the second-best results are underlined. We divide the compared methods into two classes: static multi-view learning and view incremental learning.

**Static Multi-view Learning.** The three compared methods, DUANet Geng et al. (2021), LGC-NFF Chen et al. (2023) and RCML Xu et al. (2024) are designed to simultaneously access all views, effectively leveraging inter-view correlations. DUANet excels at integrating reliable evidence by assessing the confidence level of each view. In contrast, LGCNFF and RCML achieve impressive results by capturing both consistent and complementary information from multi-view data. Notably, static multi-view learning performs exceptionally well on datasets with fewer high-quality views; however, when confronted with datasets containing a larger number of views, it struggles to capture overall cognition and accurately assess view quality.

| Classification | | Static Multi-view Learning | | | Multi-view Incremental Learning | | | | |
|---|---|---|---|---|---|---|---|---|---|
| Datasets | Metric | DUANet | LGCNFF | RCML | GAT | SI | MAS | MVCIL | HVOP |
| Animals | ACC | 68.29 (0.46) | 74.15 (1.32) | 82.11 (0.17) | 39.20 (0.36) | 45.86 (0.14) | 46.29 (0.16) | 67.03 (0.03) | **84.73 (0.12)** |
| | P | 66.63 (0.69) | 69.07 (1.71) | 78.43 (0.44) | 34.18 (1.33) | 39.92 (0.18) | 39.89 (0.28) | 64.86 (0.10) | **82.53 (0.26)** |
| | R | 61.54 (0.45) | 65.49 (1.46) | 76.03 (0.18) | 32.45 (0.44) | 38.15 (0.04) | 38.59 (0.18) | 60.53 (0.03) | **77.93 (0.03)** |
| | MAF1 | 61.43 (0.55) | 65.20 (1.59) | 76.08 (0.20) | 31.53 (0.69) | 37.03 (0.13) | 37.19 (0.02) | 61.05 (0.00) | **78.43 (0.03)** |
| Flower17 | ACC | 58.24 (1.24) | 45.35 (4.20) | 30.42 (1.92)) | 17.78 (1.56) | 44.81 (0.20) | 47.10 (0.20) | 40.93 (1.63) | **69.34 (0.42)** |
| | P | 60.22 (1.75) | 55.44 (6.80) | 35.06 (4.12) | 10.53 (3.75) | 43.26 (0.22) | 45.82 (0.33) | 42.01 (1.62) | **70.29 (0.39)** |
| | R | 58.24 (1.24) | 45.32 (4.20) | 30.42 (1.92) | 17.78 (1.56) | 44.81 (0.20) | 47.10 (0.20) | 40.93 (1.63) | **69.34 (0.42)** |
| | MAF1 | 57.31 (1.13) | 39.75 (5.54) | 25.06 (2.52) | 10.19 (2.77) | 42.86 (0.22) | 45.61 (0.31) | 38.53 (2.18) | **69.10 (0.56)** |
| Iaprtc12 | ACC | 37.53 (0.88) | 57.84 (1.27) | 52.36 (1.34) | 34.98 (0.12) | 57.00 (0.10) | 57.01 (0.10) | 43.12 (0.22) | **64.59 (0.30)** |
| | P | 42.69 (1.21) | 60.18 (1.47) | **66.83 (0.77)** | 54.52 (0.18) | 57.94 (0.09) | 57.95 (0.09) | 42.10 (0.32) | 64.65 (0.41) |
| | R | 37.29 (0.89) | 58.87 (1.41) | 51.65 (1.64) | 31.08 (0.24) | 57.81 (0.10) | 57.82 (0.10) | 44.18 (0.19) | **66.64 (0.24)** |
| | MAF1 | 37.83 (0.97) | 59.28 (1.39) | 55.22 (1.48) | 27.69 (0.33) | 57.85 (0.10) | 57.86 (0.10) | 42.14 (0.21) | **65.23 (0.35)** |
| NGs | ACC | 32.62 (2.66) | 90.65 (1.27) | 81.53 (0.58) | 72.52 (4.74) | 79.11 (1.33) | 83.56 (0.67) | 58.89 (7.11) | **95.19 (0.10)** |
| | P | 57.28 (4.22) | 92.02 (0.66) | 87.01 (0.29) | 75.00 (4.66) | 82.56 (0.07) | 84.13 (0.62) | 60.38 (8.56) | **95.40 (0.08)** |
| | R | 32.62 (2.66) | 90.67 (1.27) | 81.53 (0.58) | 72.52 (4.74) | 79.11 (1.33) | 83.56 (0.67) | 58.89 (7.11) | **95.19 (0.10)** |
| | MAF1 | 25.84 (3.66) | 90.57 (1.23) | 82.32 (0.56) | 72.50 (5.12) | 79.31 (1.30) | 83.60 (0.64) | 58.81 (7.67) | **95.21 (0.11)** |
| NoisyMNIST_15000 | ACC | 73.28 (2.86) | 89.78 (0.48) | 86.81 (0.11) | 73.43 (0.61) | 82.74 (2.10) | 90.02 (0.29) | 90.24 (0.62) | **90.79 (0.47)** |
| | P | 72.85 (4.46) | 89.75 (0.47) | 86.89 (0.08) | 74.02 (0.87) | 85.44 (0.97) | 90.20 (0.45) | 90.26 (0.46) | **90.64 (0.50)** |
| | R | 72.92 (2.81) | 89.58 (0.49) | 86.37 (0.11) | 72.89 (0.63) | 82.45 (2.07) | 89.80 (0.30) | 90.04 (0.47) | **90.60 (0.64)** |
| | MAF1 | 72.14 (3.87) | 89.58 (0.49) | 86.30 (0.12) | 72.43 (0.71) | 82.25 (1.96) | 89.83 (0.28) | 90.00 (0.48) | **90.59 (0.72)** |
| YaleB_Extended | ACC | **66.57 (1.81)** | 34.01 (1.43) | 62.59 (0.57) | 31.19 (0.82) | 32.76 (0.25) | 32.86 (0.16) | 64.16 (0.82) | 64.53 (0.55) |
| | P | 74.48 (1.91) | 48.32 (6.29) | **81.25 (0.84)** | 60.14 (4.80) | 36.12 (0.33) | 36.98 (0.45) | 67.45 (0.97) | 67.88 (0.23) |
| | R | **66.54 (1.79)** | 33.99 (1.40) | 62.61 (0.56) | 31.21 (0.86) | 32.83 (0.25) | 32.92 (0.15) | 64.24 (0.82) | 64.59 (0.55) |
| | MAF1 | **67.62 (1.82)** | 33.24 (3.38) | 67.22 (0.39) | 34.01 (1.87) | 33.51 (0.23) | 33.85 (0.00) | 64.86 (0.81) | 65.15 (0.51) |

Table 1: Node classification performance with various algorithms. Among them, bold represents the optimal value, and underline represents the suboptimal value.

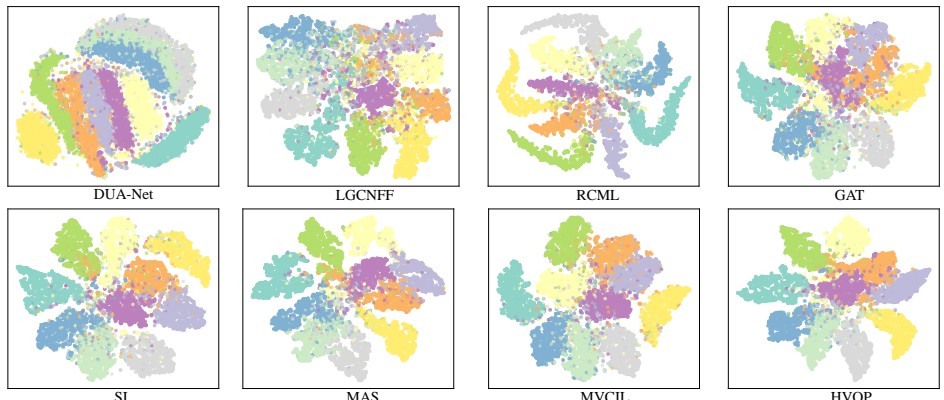

Figure 3: The visualization for multi-view semi-supervised classification on NoisyMNIST_15000.

**Multi-view Incremental Learning.** First, we incorporate GAT Veličković et al. (2018) into the view incremental framework and observe that GAT struggles to retain information from previous views, with its performance largely dependent on the quality of the most recent view data rather than the integration of all views. We then compare this to continual learning methods that leverage synaptic plasticity, such as SI Zenke et al. (2017) and MAS et al. (2018), which aim to capture overall information through plasticity mechanisms. However, these methods face challenges when dealing with low-quality views and large category sets. Additionally, we compare our approach to multi-view class-incremental learning (MVCIL) methods Li et al. (2024a). MVCIL demonstrates strong performance in handling datasets with multiple views, largely due to its effective retention of previously learned knowledge. From the Table 1, it demonstrates that even with dynamic multi-view data, our model effectively captures a comprehensive representation as the number of views increases. Moreover, it reinforces the model's ability to extract maximal information from each view while seamlessly and efficiently integrating the continuously expanding view data. We draw t-SNE visualizations for node representations in Fig. 3.

## 4.2 STABILITY IN KNOWLEDGE RETENTION: DEMONSTRATION OF VIEW FORGETTING RELIEF (EQ2)

To confirm that HVOP excelled at memorizing and integrating knowledge of old views, we tested it on inductive reasoning in Fig. 4. We evaluated the performance of GCN and HVOP under view incrementl settings on previous views. The results show that GCN experiences more pronounced degradation in view performance, whereas HVOP demonstrates a smoother decline, even stabilizing in some instances. This suggests that the HVOP model has a strong capacity for memory retention, maintaining stable knowledge of past views while fostering the development of a more comprehensive overall cognition, even as new views are continuously introduced.

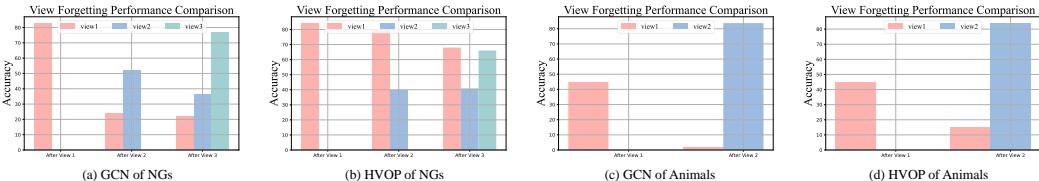

(a) GCN of NGs          (b) HVOP of NGs          (c) GCN of Animals          (d) HVOP of Animals

Figure 4: A figure illustrates the phenomenon of view forgetting for HVOP vs. GCN in scenarios with incremental views.

## 4.3 MITIGATING FORGETTING: VERIFICATION OF KNOWLEDGE TRANSFER ABILITY (EQ3)

We plot the change in performance as the views continue to increase, and add the results of training with only one view as a comparison, on the Fig. 5. We observed that the performance of single-view learning is inferior to that of HVOP, indicating that HVOP effectively facilitates knowledge transfer and integration between views during the learning process, thereby enhancing overall cognitive ability. This finding not only validates the effectiveness of the multi-view incremental strategy, but also highlights its clear superiority over the traditional single-view learning paradigm. By integrating both coherent and complementary information from multiple views, model gains a more comprehensive understanding of data, resulting in more accurate and robust predictions or classifications.

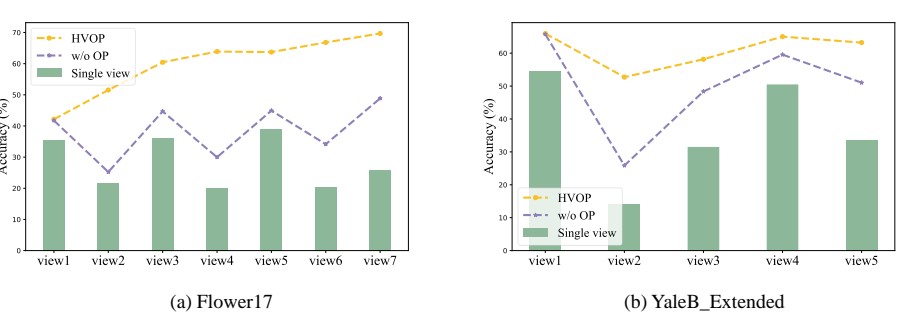

(a) Flower17          (b) YaleB_Extended

Figure 5: A comparison of HVOP streaming input and single-view learning performance.

More further, we carefully designed the ablation experiment by removing the orthogonal projection module (shown as the purple folded line in Fig. 5) to visualize its importance. The experimental results show that the performance is significantly affected by the fluctuations in the quality of the new view data, showing an unstable behavior. This finding not only highlights the critical role of the orthogonal projection module in stabilising the model performance, but also strongly validates the role of the forgetting problem in constraining the ability to transfer knowledge. If the forgetting phenomenon is not effectively mitigated, the model will have difficulty in absorbing new knowledge while maintaining a solid memory of the old knowledge, thus leading to a significant reduction.

It highlights the importance of paying great attention to the forgetting problem when constructing multi-view learning models, and exploring effective strategies to maintain the durability and accuracy of model memory to ensure smooth and efficient knowledge transfer. While our HVOP not only

underscores the model's exceptional capacity for knowledge accumulation and integration, but also demonstrates its ability to continuously absorb and utilize new knowledge from additional views while preserving previously learned information.

### 4.4 Stability in Learning: Convergence Insights (EQ3)

In Fig. 6(a), we show the loss convergence of the NGs dataset under both GCN and HVOP methods. Each view data was trained for 800 rounds. It can be clearly seen that the loss of GCN fluctuates sharply when the new view data comes, indicating that knowledge transfer and integration are not carried out effectively. In contrast, under the HVOP framework, the change of view data has less impact on the loss, which indicates that HVOP can well integrate the association between the old and new data, and promote the overall cognition to progress continuously.

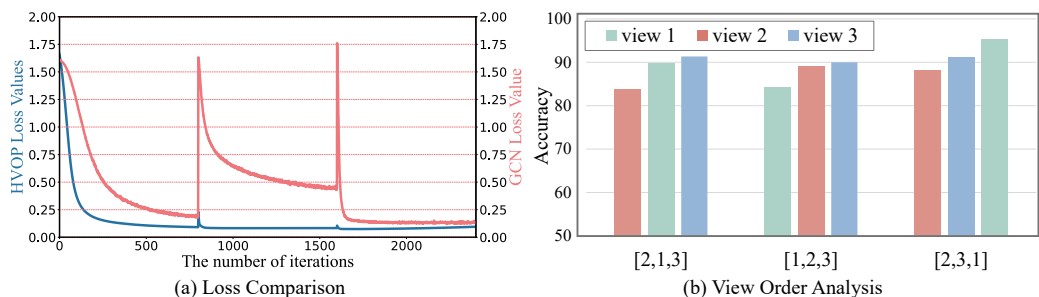

(a) Loss Comparison        (b) View Order Analysis

Figure 6: (a) Comparison of the loss reduction between HVOP and GCN in learning NGs dataset. (b) The overall performance of learning from NGs datasets in different view orders.

### 4.5 Cognitive Pathways Investigate: View Order Analysis (EQ3)

In Fig. 6(b), we present the impact of different sequential combinations of views on HVOP. Unsurprisingly, all variations of view permutations had a positive effect on the final overall cognition. While the diversity of these combinations resulted in distinct cognitive paths, they consistently pointed to a common trend—a steady improvement in overall cognitive ability. In future work, we will further explore the influence of sequencing on overall cognition.

## 5 Discussions and Conclusions

The traditional multi-view learning frameworks are often limited by their ability to handle static multi-view data, making it difficult to adapt to the dynamic growth and changes in real-world view data. In response, this study proposes a Hebbian-based View Orthogonal Projection framework, called HVOP to overcome this challenge. Unlike continual learning, which focuses on adapting to sequential tasks, HVOP emphasizes view incremental learning, aiming to maintain the stability of the overall cognitive structure while focusing on the in-depth understanding of a single view in each learning task. We have thoroughly analyzed the phenomenon of view forgetting, which refers to the issue of forgetting old view information when learning new views, and for the first time, as view transfer learning. This concept hypothesizes that by effectively mitigating view forgetting, it is possible to achieve smooth transfer and integration of knowledge across views. To realize this goal, the HVOP method cleverly integrates recursive lateral connection mechanisms with Hebbian learning principles. This combination not only facilitates efficient knowledge extraction across views but also ensures deep integration and continual optimization of both new and old view information within the neural network. The experimental results exhibited that HVOP significantly enhances the model's adaptability and generalization ability in handling dynamic multi-view data.

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
