# OpenReview forum: "Brain-inspired Multi-View Incremental Learning for Knowledge Transfer and Retention"
_ICLR.cc/2025/Conference — ICLR 2025 Conference Withdrawn Submission_

### Official Review · Reviewer_E8Bv · 2024-10-22

**Soundness:** 2
**Presentation:** 2
**Contribution:** 2
**Rating:** 3
**Confidence:** 3

**Summary:**

This work focuses on flexible memory retention and efficient transfer for continual learning across multiple views. By proposing the HVOP, the authors address the forgetting problem, establishing a more robust framework for dynamically and incrementally available data. The empirical results demonstrate various metrics alongside visual analysis of the HVOP-learned representations based on datasets such as Flower and YaleB.

**Strengths:**

1.The cognitive science perspective on the knowledge transfer problem is interesting, drawing connections with human memory retention.

2.The literature review is sufficient, and the experiments are well-conducted with proper baselines.

**Weaknesses:**

1. The presentation of the paper needs enhancement, as the technical contributions are somewhat ambiguous.
2. The experiments lack adequate evaluation, with insufficient metrics and discussion.

**Questions:**

1. The forgetting problem has been a longstanding challenge in continual learning. How does the proposed method relate to similar observations in transfer learning and continual learning? While the definitions and discussions in the methodology are promising, they could be more closely tailored to the specific problems addressed in this work.
2. How does KTS differ from SI and other methods that rely on additional guidance for parameter update gradient directions? The novelty of the approach is not clearly articulated; instead, there is a sequential introduction of methods, such as Hebbian learning. Proper citations and clarifications are needed.
3. The experiments lack basic metrics, such as forgetting rates and parameter costs, which are crucial for the tasks considered. Most evaluations focus on classification tasks. Although the latent space visualization is interesting, the discussion on L. 431 does not extend further, leaving conclusions unaddressed.
4. More conclusions from the performance of HVOP regarding memory retention and transfer in the hippocampus could be discussed, since the main motivation is to draw a connection between the biological model of human memory systems.

---

### Official Review · Reviewer_Uncu · 2024-11-01

**Soundness:** 3
**Presentation:** 3
**Contribution:** 3
**Rating:** 6
**Confidence:** 3

**Summary:**

The paper presents a biologically inspired framework, Hebbian View Orthogonal Projection (HVOP), to address the challenges of knowledge retention and transfer in multi-view incremental learning. This work is inspired by neural mechanisms such as Hebbian learning and recursive lateral connections. Orthogonal projection is employed to reduce interference between old and new views. Experiments on node classification tasks demonstrate the HVOP outperforms traditional methods.

**Strengths:**

This paper makes contributions by addressing the critical issue of knowledge retention and transferring in multi-view learning through a biologically inspired framework.

Its originality lies in the application of Hebbian learning and lateral connections, inspired by the brain’s ability.

Despite the heavy focus on biology, this paper provides clear presentation and enables readers to follow the biological analogies easily.

Extensive experiments are performed to demonstrate the effectiveness of the proposed approach.

**Weaknesses:**

More intuition or visual explanations about how the recursive lateral connections operate within the model can help improve the presentation.

What is HVOP’s scalability across different data context?

What are the potential limitations of HVOP in handling high-dimensional, low-quality views?

**Questions:**

Is HVOP’s performance affected by the dimensionality of the views?

How does HVOP handle high-dimensional noisy views without experiencing significant degradation in knowledge transfer and retention?

What about the potential extension of HVOP to non-graph-based data?

---

### Official Review · Reviewer_CMTR · 2024-11-02

**Soundness:** 2
**Presentation:** 3
**Contribution:** 1
**Rating:** 3
**Confidence:** 2

**Summary:**

This paper aims to solve Multi-View Incremental Learning for Knowledge Transfer and Retention in Brain-inspired way. This paper proposes a view transfer learning framework named Hebbian View Orthogonal Projection (HVOP), which includes a knowledge transfer space (KTS), recursive lateral connections and Hebbian learning.  The proposed method endows the learning process with brain-like dynamic adaptability, enhancing knowledge transfer and integration, which are verified by experiment.

**Strengths:**

1.  Addressing incremental learning through brain-inspired methods is a promising avenue of research.

2. The paper is clearly written and includes helpful figures for improved understanding.

**Weaknesses:**

1.	The innovation of the proposed method‘s design is difficult to assess due to a lack of detailed comparisons with other brain-inspired methods or similar approaches.

2.	The experimental design is unclear, as the paper does not provide information on the training setup or details about the dataset.

3.	Additional experiments on a larger dataset, such as Mini-ImageNet, are necessary to fully evaluate the potential and limitations of the proposed method.

**Questions:**

N/A

---

### Official Review · Reviewer_92pk · 2024-11-03

**Soundness:** 2
**Presentation:** 2
**Contribution:** 2
**Rating:** 3
**Confidence:** 4

**Summary:**

This paper introduces the Hebbian View Orthogonal Projection (HVOP) model, aiming to tackle the view-forgetting problem in multi-view incremental learning. Inspired by brain-like mechanisms, HVOP employs orthogonal projection and recursive lateral connections to promote knowledge retention and transfer across different views. The authors present results on node classification tasks, claiming HVOP’s superior performance in maintaining previously acquired knowledge while adapting to new information.

**Strengths:**

1. This paper presents a novel framework (HVOP) for multi-view incremental learning, inspired by biological mechanisms, specifically synaptic plasticity and orthogonal projection, to address the persistent issue of view-forgetting.
2. The authors attempt to address a significant limitation in traditional multi-view learning by designing a model to maintain knowledge retention and facilitate transfer between views.
3. The model introduces a Knowledge Transfer Space (KTS) aimed at reducing interference between old and new views, which could significantly enhance the retention of prior knowledge and integration of new information. If improved, this framework could advance multi-view learning methods, particularly in scenarios with evolving data.

**Weaknesses:**

1. Ambiguity in Mathematical Explanations and Formulae: The mathematical formulation in the methodology lacks sufficient clarity, which impedes understanding of both the model’s operation and its novelty.

	(1) Key symbols in core equations are not adequately explained. In Eq. (1), σ and D ̃ are introduced without definition, making it difficult for readers to grasp the exact transformations applied.

	(2) Eq. (3) is presented without clearly explaining its function within the model. The subsequent phrase "To address this" suggests that the feasibility of this equation needs to be ensured, yet the specified KTS does not clarify how Eq. (3) is achieved. This lack of explanation limits the reader’s ability to follow the authors' logic and assess the novelty and contributions of the model.

	(3) The methodology section introduces several terms and equations without clarifying their relationships (e.g., Eq. 3 and Eq. 4 for orthogonal projection).

2. Ambiguity in Addressing the View Forgetting Problem: Section 4.2 demonstrates HVOP’s effectiveness in combating view forgetting. However, the results suggest that this may come at the cost of reduced accuracy on new views. Further discussion on this potential trade-off would strengthen the paper’s contributions by clarifying whether HVOP sacrifices new-view performance for retention. Additionally, Figure 4’s font size should be adjusted for improved readability.

3. Insufficient Analysis of Pictures: For Fig. 6(a), the objects represented by the two types of lines are not indicated; it is recommended to use a direct legend. Additionally, the paper lacks analysis of the results shown in Fig. 3. Moreover, the reference to Fig. 2 in line 82 should be corrected to Fig. 1.

**Questions:**

1. Could the authors provide clearer definitions for symbols and terms in key equations, particularly in Eqs. (1) and (3)?
2. Does HVOP sacrifice new-view accuracy to enhance the retention of old views, as suggested by Section 4.2? A detailed analysis of this trade-off would be valuable.
3. Could the authors further analyze why Fig. 3 demonstrates HVOP’s advantages?
4. How does HVOP compare with baseline models regarding computational efficiency, given the added complexity of recursive lateral connections?

---

### Note · Authors · 2024-11-22

I have read and agree with the venue's withdrawal policy on behalf of myself and my co-authors.